# PREDICTING VIDEO WITH VQVAE

## ABSTRACT

In recent years, the task of video prediction—forecasting future video given past video frames—has attracted attention in the research community. In this paper we propose a novel approach to this problem with Vector Quantized Variational AutoEncoders (VQ-VAE). With VQ-VAE we compress high-resolution videos into a hierarchical set of multi-scale discrete latent variables. Compared to pixels, this compressed latent space has dramatically reduced dimensionality, allowing us to apply scalable autoregressive generative models to predict video. In contrast to previous work that has largely emphasized highly constrained datasets, we focus on very diverse, large-scale datasets such as Kinetics-600. We predict video at a higher resolution, $256 \times 256$, than any other previous method to our knowledge. We further validate our approach against prior work via a crowdsourced human evaluation.

## 1 INTRODUCTION

When it comes to real-world image data, deep generative models have made substantial progress. With advances in computational efficiency and improvements in architectures, it is now feasible to generate high resolution, realistic images from vast and highly diverse datasets (Brock et al., 2019; Razavi et al., 2019; Karras et al., 2017). Apart from the domain of images, deep generative models have also shown promise in other data domains such as music (Dieleman et al., 2018; Dhariwa et al., 2020), speech synthesis (Oord et al., 2016), 3D voxels (Liu et al., 2018; Nash & Williams, 2017), and text (Radford et al., 2019). One particular fledgling domain is video.

While some work in the area of video generation (Clark et al., 2020; Vondrick et al., 2016; Saito & Saito, 2018) has explored video synthesis—generating videos with no prior frame information—many approaches actually focus on the task of video prediction conditioned on past frames (Ranzato et al., 2014; Srivastava et al., 2015; Patraucean et al., 2015; Mathieu et al., 2016; Lee et al., 2018; Babaeizadeh et al., 2018; Oliu et al., 2018; Xiong et al., 2018; Xue et al., 2016; Finn et al., 2016; Luc et al., 2020). It can be argued that video synthesis is a combination of image generation and video prediction. In other words, one could decouple the problem of video synthesis into unconditional image generation and conditional video prediction from a generated image. Therefore, we specifically focus on video prediction in this paper. Potential computer vision applications of video forecasting include interpolation, anomaly detection, and activity understanding. More generally, video prediction also has more general implications for intelligent systems—the ability to anticipate the dynamics of the environment. The problem is thus also relevant for robotics and reinforcement learning (Finn et al., 2016; Ebert et al., 2017; Oh et al., 2015; Ha & Schmidhuber, 2018; Racanire et al., 2017).

Approaches toward video prediction have largely skewed toward variations of generative adversarial networks (Mathieu et al., 2016; Lee et al., 2018; Clark et al., 2020; Vondrick et al., 2016; Luc et al., 2020). In comparison, we are aware of only a relatively small number of approaches which propose variational autoencoders (Babaeizadeh et al., 2018; Xue et al., 2016; Denton & Fergus, 2018), autoregressive models (Kalchbrenner et al., 2017; Weissenborn et al., 2020), or flow based approaches (Kumar et al., 2020). There may be a number of reasons for this situation. One is the explosion in the dimensionality of the input space. A generative model of video needs to model not only one image but tens of them in a coherent fashion. This makes it difficult to scale up such models to large datasets or high resolutions. In addition, previous work (Clark et al., 2020) suggests that video prediction may be fundamentally more difficult than video synthesis; a synthesis model can generate simple samples from the dataset while prediction potentially forces the model to forecast conditioned on videos that are outliers in the distribution. Furthermore, most prior work has focused on datasets with low scene

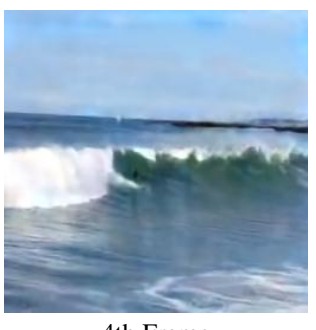 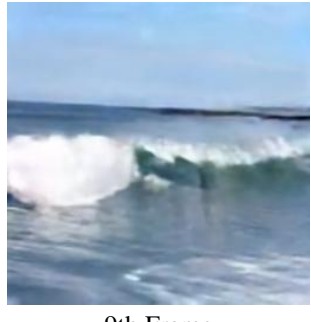 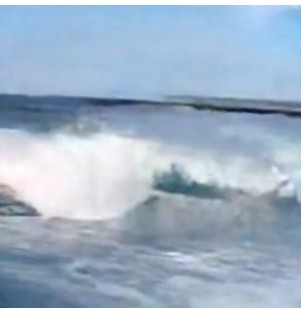

| 4th Frame | 9th Frame | 16th Frame |

Figure 1: In this paper we predict video at a high resolution ($256 \times 256$) using a compressed latent representation. The first 4 frames are given as conditioning. We predict the next 12, two of which (9th and 16th) we show on the right. All frames shown here have been compressed by VQ-VAE. Videos licensed under CC-BY. Attribution for videos in this paper can be found in supplementary material. Best seen in video in the supplementary material.

diversity such as Moving MNIST (Srivastava et al., 2015), KTH (Schuldt et al., 2004), or robotic arm datasets (Finn et al., 2016; Ebert et al., 2017). While there have been attempts to *synthesize* video at a high resolution (Clark et al., 2020), we know of no attempt—excluding flow based approaches—to *predict* video beyond resolutions of 64x64.

In this paper we address the large dimensionality of video data through compression. Using Vector Quantized Variational Autoencoders (VQ-VAE) (van den Oord et al., 2017), we can compress video into a space requiring only 1.3% of the bits expressed in pixels. While this compressed encoding is lossy, we can still reconstruct the original video from the latent representation with a high degree of fidelity. Furthermore, we can leverage the modularity of VQ-VAE and decompose our latent representation into a hierarchy of encodings, separating high-level, global information from details such as fine texture or small motions. Instead of training a generative model directly on pixel space, we can instead model this much more tractable discrete representation, allowing us to train much more powerful models, use large diverse datasets, and generate at a high resolution. While most prior work has focused on GANs, this discrete representation can also be modeled by likelihood-based models. Likelihood models in concept do not suffer from mode-collapse, instability in training, and lack of diversity of samples often witnessed in GANs (Denton & Fergus, 2018; Babaeizadeh et al., 2018; Razavi et al., 2019). In this paper, we propose a PixelCNN augmented with causal convolutions in time and spatiotemporal self-attention to model this space of latents. In addition, because the latent representation is decomposed into a hierarchy, we can exploit this decomposition and train separate specialized models at different levels of the hierarchy.

Our paper makes four contributions. First, we demonstrate the novel application of VQ-VAE to video data. Second, we propose a set of spatiotemporal PixelCNNs to predict video by utilizing the latent representation learned with VQ-VAE. Third, we explicitly predict video at a higher resolution than ever before. Finally, we demonstrate the competitive performance of our model with a crowdsourced human evaluation.

## 2 BACKGROUND

### 2.1 VECTOR QUANTIZED AUTOENCODERS

VQ-VAEs (van den Oord et al., 2017) are autoencoders which learn a discrete latent encoding for input data $x$. First, the output of non-linear encoder $z_e(x)$, implemented by a neural network, is passed through a discretization bottleneck. $z_e(x)$ is mapped via nearest-neighbor into a quantized codebook $e \in R^{K \times D}$ where $D$ is the dimensionality of each vector $e_j$ and $K$ is the number of categories in the codebook. The discretized representation is thus given by:

$$z_q(x) = e_k \text{ where } k = \text{argmin}_j ||z_e(x) - e_j||_2 \qquad (1)$$

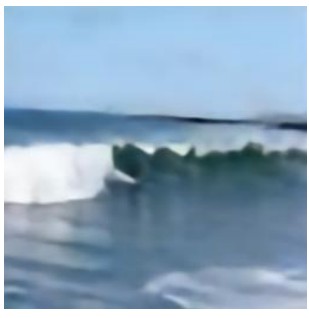 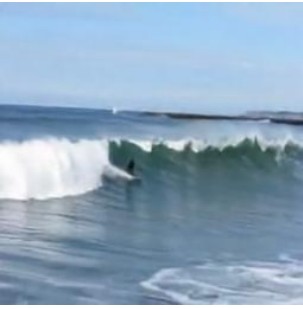 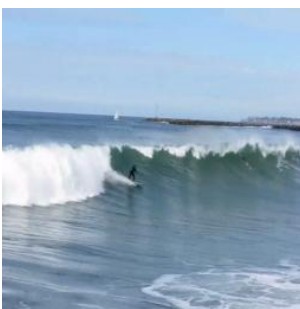

Top Layer          Top + Bottom Layer          Original Frame

Figure 2: Here we demonstrate the compression capability of VQ-VAE. The top and bottom rows represent two different frames within the same video. The top layer retains most of the global information, while the bottom layer adds fine detail. Videos licensed under CC-BY. Attribution for videos in this paper can be found in the supplementary material

Equation 1 is not differentiable; however, (van den Oord et al., 2017) notes that copying the gradient of $z_q(x)$ to $z_e(x)$ is a suitable approximation similar to the straight-through estimator (Bengio et al., 2013). A decoder $D$, also implemented by a neural network, then reconstructs the input from $z_q(x)$. The total loss function for the VQ-VAE is thus:

$$L = ||D(z_g(x)) - x||_2^2 + ||\text{sg}[z_g(x)] - \mathbf{e}||_2^2 + \beta||z_g(x) - \text{sg}[\mathbf{e}]||_2^2 \quad (2)$$

Where sg is a stop gradient operator, and $\beta$ is a parameter which regulates the rate of code change. As in previous work (van den Oord et al., 2017; Razavi et al., 2019), we replace the second term in equation 2 and learn the codebook $e \in R^{K \times D}$ via an exponential moving average of previous values during training:

$$N_i^{(t)} := N_i^{(t-1)} * \gamma + n_i^{(t)}(1 - \gamma), \; m_i^{(t)} := m_i^{(t-1)} * \gamma + \sum_j^{n_i^{(t)}} z_e(x)_{i,j}^{(t)}(1 - \gamma), \; e_i^{(t)} := \frac{m_i^{(t)}}{N_i^{(t)}} \quad (3)$$

Where $\gamma$ is a decay parameter and $n_i^{(t)}$ is the numbers of vectors in $z_g(x)$ in a batch that will map to $e_i$.

## 2.2 PixelCNN Models

PixelCNN and related models have shown promise in modeling a wide variety of data domains (van den Oord et al., 2016; Oord et al., 2016; Kalchbrenner et al., 2017; Weissenborn et al., 2020). These autoregressive models are likelihood-based—they explicitly optimize negative log-likelihood. They exploit the fact that the joint probability distribution input data $x$ can be factored into a product of conditional distributions for each dimension of the data:

$$P_\theta(\mathbf{x}) = \prod_{i=0}^{n} p_\theta(x_i|x_{<i}) \quad (4)$$

Where $n$ is the full dimensionality of the data. This factorization is implemented by a neural network, and the exact set of conditional dependencies is determined by the data domain. Image pixels may depend on regions above and to the left of them (van den Oord et al., 2016), while temporal dimensions may depend on past dimensions (Oord et al., 2016; Kalchbrenner et al., 2017; Weissenborn et al., 2020).

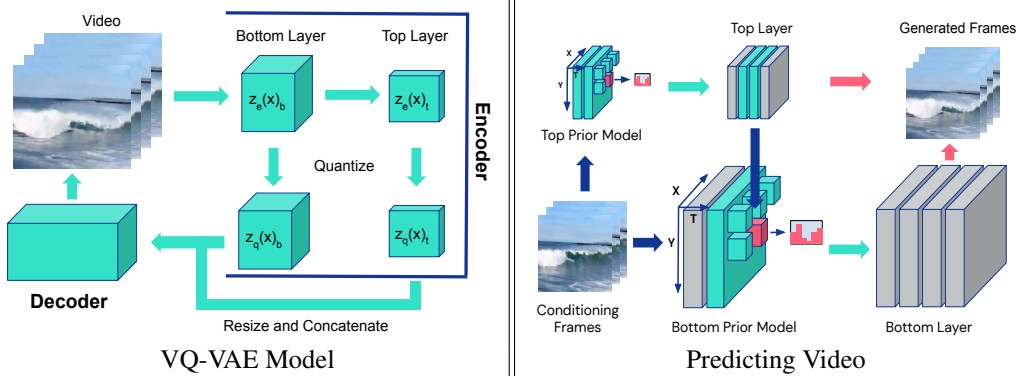

Figure 3: Here we show an overview of our approach. On the left we show the process of compressing video with VQ-VAE. On the right we show the process of generating video with the latents. The top conditional prior model is a PixelCNN with causal convolutions to incorporate all past information at each point in space-time. The bottom conditional prior model is simply a 2D PixelCNN which generates slice by slice. It is conditioned with a convolutional tower which incorporates a window of time slices from the top latents and past bottom latents. The slices outside of this window are colored grey in this diagram. Blue arrows represent conditioning, green arrows generation, and pink feed-forward decoding. Videos licensed under CC-BY. Attribution for videos in this paper can be found in the supplementary material.

## 3 METHOD

Our approach consists of two main components. First, we compress video segments into a discrete latent representation using a hierarchical VQ-VAE. We then propose a multi-stage autoregressive model based on the PixelCNN architecture, exploiting the low dimensionality of the compressed latent space and the hierarchy of the representation.

### 3.1 COMPRESSING VIDEO WITH VQ-VAE

Similar to (Razavi et al., 2019), we use VQ-VAE to compress video in a hierarchical fashion. This multi-stage composition of the latent representation allows decomposition of global, high level information from low-level details such as edges or fine motion. For image information (Razavi et al., 2019) this approach confers a number of advantages. First, the decomposition allows latent codes to specialize at each level. High level information can be represented in an even more compressed manner, and the total reconstruction error is lower. In addition, this hierarchy leads to a naturally modular generative model. We can then develop a generative model that specializes in modeling the high-level, global information. We can then train a separate model, conditioned on global information, that fills in the details and models the low-level information further down the hierarchy. In this paper, we adopt the terminology of (Razavi et al., 2019) and call the set of high-level latents the *top* layer and the low-level latents the *bottom* layer.

Consistent with the experimental setup of previous work in video prediction, we deal with 16-frame videos. Most of the videos in our training dataset are 25 frames per second. We use frames at a $256 \times 256$ resolution. The full video voxel is thus $256 \times 256 \times 16$. Using residual blocks with 3D convolutions, we downsample the video spatiotemporally. At the bottom layer, the video is downsampled to a quantized latent space of $64 \times 64 \times 8$, reducing the spatial dimension by 4 and the temporal dimension by 2. Another stack of blocks reduces all dimensions by 2, with a top layer of $32 \times 32 \times 4$. Each of the voxels in the layer is quantized into 512 codes with a different codebook for both layers.

The decoder then concatenates the bottom layer and the top layer after upsampling using transposed convolutions. From this concatenation as input, the decoder deterministically outputs the full $256 \times 256 \times 16$ video. Overall, we reduce a $256 \times 256 \times 16 \times 3 \times \log(256)$ space down to a $64 \times 64 \times 8 \times \log(512) + 32 \times 32 \times 4 \times \log(512)$ space, a greater than 98% reduction in bits required.

Table 1: Human Evaluation on Kinetics-600

| Prefer Video VQ-VAE | Prefer (Luc et al., 2020) | Indifferent |
|---|---|---|
| 65.7% | 12.8% | 21.5% |

During training, we randomly mask out the bottom layer in the concatenated input to the decoder. Masking encourages the model to utilize the top latent layer and prevent codebook collapse.

## 3.2 Predicting Video with PixelCNNs

With VQ-VAE, our $256 \times 256 \times 16$ video is now decomposed into a hierarchy of quantized latents at $64 \times 64 \times 8$ and $32 \times 32 \times 4$. Previous autoregressive approaches involved full pixel videos at $64 \times 64 \times 16$ (Weissenborn et al., 2020) or $64 \times 64 \times 20$ (Kalchbrenner et al., 2017). Our latent representation is thus well within the range of tractability for these models. Furthermore, given the hierarchy, we can factorize our generative model into a coarse-to-fine fashion. We denote the model of the top layer the *top prior* and model of the bottom layer the *bottom prior*. Because we are focusing on video prediction, we emphasize that both are still conditioned on a series of input frames. While previous work used 5 frames and predicted 11 (Weissenborn et al., 2020; Clark et al., 2020), the power-of-two design of our architecture leads us to condition on 4 and predict 12. When conditioning our prior models on these frames, we need not use a large stack directly on the original images but save memory and computation by training a smaller stack of residual layers on their latent representation, compressing these 4 conditional frames into a small latent space of $32 \times 32$ and $64 \times 64 \times 2$.

We first model the top layer with a conditional prior model. Our prior model is based on a PixelCNN with multi-head self attention layers (Chen et al., 2018). We adapt this architecture by extending the PixelCNN into time; instead of a convolutional stack over a square image, we use a comparable 3D convolutional stack over the cube representing the prior latents. The convolutions are masked in the same way as the original PixelCNN in space—at each location in space, the convolutions only have access to information to the left and above them. In time, present and future timesteps are masked out, and convolutions only have access to previous timesteps. While spatiotemporal convolutions can be resource intensive, we can implement most of this functionality with 2D convolutions separately in the $x - t$ plane and the $y - t$ plane. We take 1D convolutions in the horizontal and vertical stacks in the original PixelCNN (van den Oord et al., 2016) and add the extra dimension of time to them, making them 2D. Our only true 3D convolution is at the first layer before the addition of gated stacks. We use multi-head attention layers analogous to (Razavi et al., 2019); this time the attention is applied to a 3D voxel instead of a 2D layer as in (Razavi et al., 2019). Attention is applied every five layers. During sampling we can generate voxels left-to-right, top-to-bottom within each temporal step as in the original PixelCNN. Once a final step is generated, we can generate the next step conditioned on the previous generated steps.

Once we have a set of latents from the top layer, we can condition our bottom conditional prior model and generate the final bottom layer. Because the bottom layer has a higher number of dimensions and relies on local information, we don't necessarily need a 3D PixelCNN. Instead, we use a 2D PixelCNN with multi-head self attention every five layers analogous to (Razavi et al., 2019). We implement a 3D conditional stack, however, that takes in a window of time steps from the top layer as well as a window of past generated time steps in the bottom layer. The window sizes we used were 4 and 2 respectively. This conditional stack is used as conditioning to the 2D PixelCNN at the current timestep.

## 4 Related Work

**Video Prediction and Synthesis:**  In the last few years, the research community has focused a spotlight on the topic of video generation—either in the form of video synthesis or prediction. Early approaches involved direct, deterministic pixel prediction (Ranzato et al., 2014; Srivastava et al., 2015; Oh et al., 2015; Patraucean et al., 2015). Given the temporal nature of video, such approaches often incorporated LSTMs. These papers usually applied their deterministic models on datasets such

Table 2: FVD Scores on Kinetics-600. Lower is better.

| Method | FVD Score ($\downarrow$) |
| --- | --- |
| Video Transformer ($64 \times 64$) (Weissenborn et al., 2020) | $170 \pm 5$ |
| DVD-GAN-FP ($64 \times 64$) (Clark et al., 2020) | $69.15 \pm 1.16$ |
| TRIVD-GAN-FP ($64 \times 64$) (Luc et al., 2020) | $25.74 \pm 0.66$ |
| Video VQ-VAE ($64 \times 64$) | $64.30 \pm 2.04$ |
| Video VQ-VAE FVD* ($64 \times 64$) | $54.30 \pm 3.49$ |
| Video VQ-VAE ($256 \times 256$) | $129.85 \pm 1.64$ |
| Video VQ-VAE FVD* ($256 \times 256$) | $82.45 \pm 1.16$ |

as moving MNIST characters (Srivastava et al., 2015); because of their deterministic nature, rarely were they successfully applied to more complex datasets. Given this situation, researchers started to adapt popular models for image generation to the problem starting with generative adversarial models (Mathieu et al., 2016; Vondrick et al., 2016; Lee et al., 2018; Babaeizadeh et al., 2018; Clark et al., 2020; Saito & Saito, 2018; Luc et al., 2020; Xiong et al., 2018), variational autoencoders (Xue et al., 2016), and autoregressive models (Kalchbrenner et al., 2017; Weissenborn et al., 2020). Others stepped aside from the problem of full pixel prediction and instead predicted pixel motion (Finn et al., 2016; Walker et al., 2016; Jia et al., 2016) or a decomposition of pixels and motion (Denton & Fergus, 2018; Gao et al., 2019; Jang et al., 2018; Hao et al., 2018; Li et al., 2018a; Tulyakov et al., 2018; Villegas et al., 2017a). Finally, some have proposed a hierarchical approach based on structured information—generating video conditioned on text (Li et al., 2018b), semantic segments (Luc et al., 2017; 2018), or human pose (Walker et al., 2017; Villegas et al., 2017b)

**Compressing Data with Latents:** The key element in our video prediction framework is compression—representing videos through lower dimensional latents. We apply the framework of VQ-VAE (van den Oord et al., 2017; Razavi et al., 2019) which has been successfully applied to compress image and sound data. Related to VQ-VAE, other researchers have explored hierarchies of latents for generation of images (Fauw et al., 2019) and music (Dieleman et al., 2018).

**Autoregressive Models:** The foundation of our model is based on PixelCNN (van den Oord et al., 2016). Distinct from implicit likelihood models such as GANs and approximate methods such as VAEs, the family of PixelCNN architectures have shown promise in modeling a variety of data domains including images (van den Oord et al., 2016), sound (Oord et al., 2016), and video (Kalchbrenner et al., 2017; Weissenborn et al., 2020). In line with our paper, recent work with these models has shifted toward decomposing autoregression through hierarchies (Menick & Kalchbrenner, 2019; Reed et al., 2017) and latent compression (van den Oord et al., 2017; Razavi et al., 2019; Dhariwa et al., 2020).

## 5 EXPERIMENTS

In this section, we evaluate our model quantitively and qualitatively on the Kinetics-600 dataset (Carreira et al., 2018). This dataset of videos is very large and highly diverse, consisting of hundreds of thousands of videos selected from YouTube across 600 actions. While most previous work has focused on more constrained datasets, only a few (Clark et al., 2020; Luc et al., 2020; Weissenborn et al., 2020) have attempted to scale to larger size and complexity. We train our top and bottom models for around 1000000 iterations with a total batch sizes of 512 and 32 respectively. Our VQ-VAE model was trained on a batch size of 16 for 1000000 iterations.

### 5.1 QUALITATIVE EVALUATION

While the Kinetics-600 dataset is publicly available for use, the individual videos in the dataset may not be licensed for display in an academic paper. Therefore, in this paper, we apply our model trained on Kinetics-600 to videos licensed under Creative Commons from the YFCC100m dataset (Thomee et al., 2015). In figure 4 we show some selected predictions. We find that our approach is able to

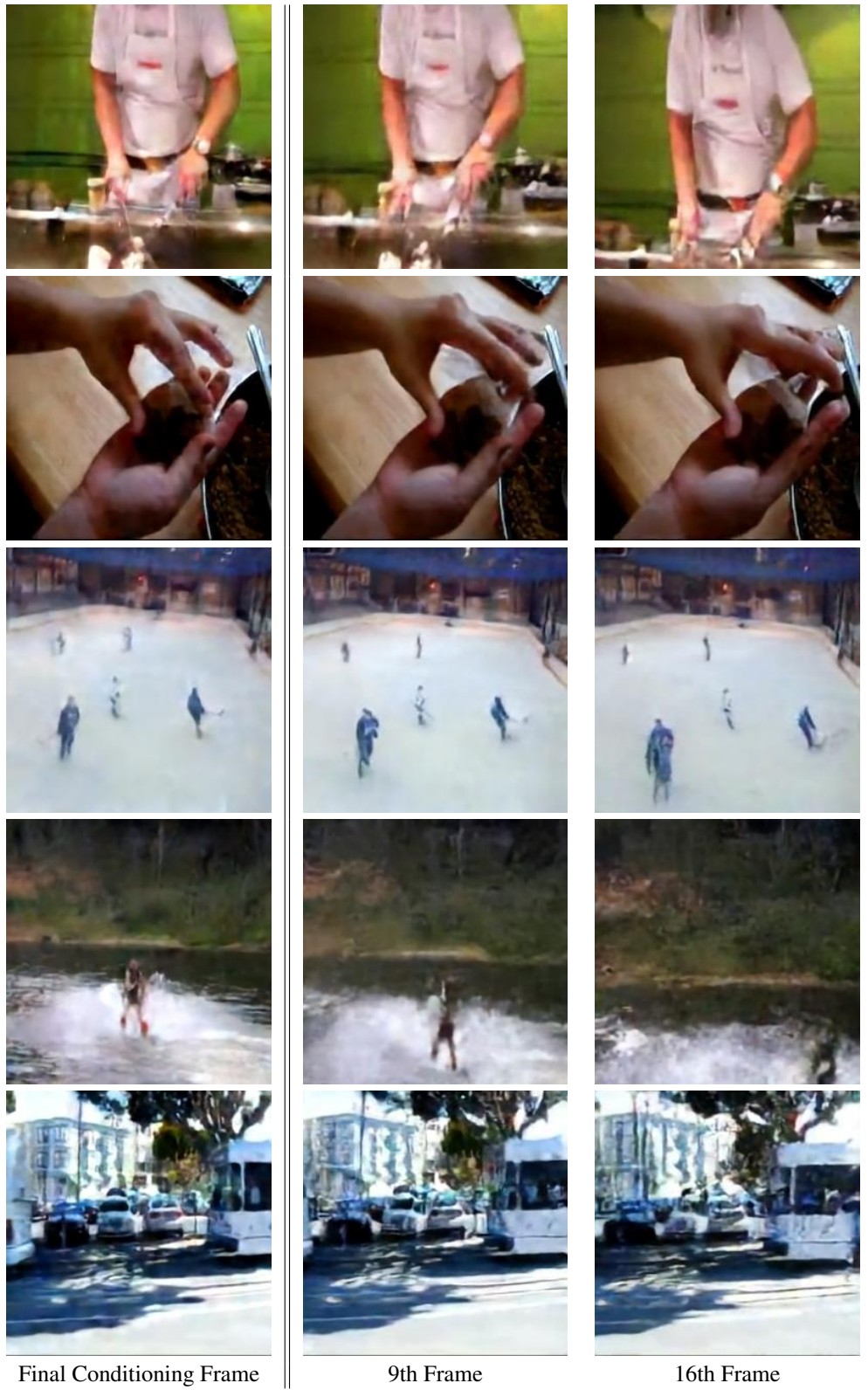

| Final Conditioning Frame | 9th Frame | 16th Frame |

Figure 4: Selected prediction results. The first 4 frames are given as conditioning. We predict the next 12, two of which (9th and 16th) we show on the right. All frames shown here have been compressed by VQ-VAE. Videos licensed under CC-BY. Attribution for videos in this paper can be found in the supplementary material. Best seen in video in the supplementary material.

model camera perspective, parallax, inpainting, deformable human motions, and even aspects of crowd motion across a variety of different visual contexts.

## 5.2 QUANTITATIVE EVALUATION

Quantitative evaluation of generative models of images, especially across different classes of models, is an open problem (Theis et al., 2016). It is even less explored in the realm of video prediction. One proposed metric used on larger datasets such as Kinetics-600 is the Fréchet Video Distance (Unterthiner et al., 2018). As no previous approach has attempted $256 \times 256$ resolution, we downscale our videos to $64 \times 64$ for a proper comparison against prior work. We also compute FVD on the full resolution as a baseline for future work. We use 2-fold cross-validation over 39000 samples to compute FVD. We show our results in Table 7. We find performance exceeds (Clark et al., 2020) but not necessarily (Luc et al., 2020). We also find that comparing the VQ-VAE samples to the reconstructions, not the original videos, leads to an even better score (shown by FVD*). This result is similar to the results on images for VQ-VAE (Razavi et al., 2019). As GAN-based approaches are explicitly trained on classifier (discriminator) based losses, FVD—a metric based on a neural-network classifier—may favor GAN-based approaches versus log-likelihood based models even if the quality of the samples are comparable. Given the possible flaws in this metric, we also conduct a human evaluation similar to (Vondrick et al., 2016). We had 15 participants compare up to 30 side-by-side videos generated from our approach and that of (Luc et al., 2020). Each video had at least 13 judgements. For each comparison, both models used the exact set of conditioning frames and had a resolution of 64x64. Participants could choose a preference for either video, or they could choose indifference—meaning the difference in quality between two videos is too close to perceive. We show our results in Table 1. Out of a total of 405 judgements, participants preferred ours 65.7% of the time, (Luc et al., 2020) 12.8%, and 21.5% were judged to be too close in quality. Even though (Luc et al., 2020) has a much lower FVD score, we find that our participants had stronger preference for samples generated from our model.

## 6 CONCLUSION

In this paper we have explored the application of VQ-VAE towards the task of video prediction. With this learned compressed space, we can utilize powerful autoregressive models to generate possible future events in video at higher resolutions. We show that we are also able to achieve a level of performance comparable to contemporary GANs.

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

# A  APPENDIX

Table 3: VQ-VAE Architecture Details

| Parameter | Value |
|---|---|
| Input Size | 256×256×16×3 |
| Latent layers | 32×32×4, 64×64×8 |
| $\beta$ (commitment loss coefficient) | 0.25 |
| Batch size | 16 |
| Hidden units | 128 |
| Residual units | 32 |
| Layers | 2 |
| Codebook size | 512 |
| Codebook dimension | 64 |
| First Stage Encoder twh-Conv Filter Size | 4 8 8 |
| First Stage Encoder twh-Conv Filter Stride | 2 4 4 |
| Second Stage Encoder twh-Conv Filter Size | 4 4 4 |
| Second Stage Encoder twh-Conv Filter Stride | 2 2 2 |
| Upsampling twh-Conv Filter Size | 4 8 8 |
| Upsampling twh-Conv Filter Stride | 2 4 4 |
| Training steps | 1000000 |

Table 4: PixelCNN Prior Details

| Parameter | Top Prior | Bottom Prior |
|---|---|---|
| Input size | $32{\times}32{\times}3$ | $64{\times}64$ |
| Batch size | 512 | 32 |
| Hidden units | 512 | 512 |
| Residual units | 1024 | 1024 |
| Layers | 40 | 20 |
| Attention layers | 8 | 4 |
| Attention heads | 8 | 8 |
| Conv Filter size | 3 | 5 |
| Dropout | 0.5 | 0.0 |
| Training steps | 1016000 | 950000 |

Table 5: Top Prior Conditioning Stack

| Parameter | Values |
|---|---|
| Input size | $32{\times}32$ (upsampled to $64{\times}64{\times}2$), $64{\times}64{\times}2$ |
| Hidden units | 512 |
| Residual units | 128 |
| Layers | 4 |

Table 6: The conditioning frames $256{\times}256{\times}4{\times}3$ are compressed using our VQ-VAE into a $32{\times}32$ and $64{\times}64{\times}2$ space. These two layers are then concatenated, downsampled back down to $32{\times}32{\times}256$ by a convolutional layer, tiled to $32{\times}32{\times}3{\times}256$, and finally fed through another convolutional layer to $32{\times}32{\times}3{\times}512$ before being fed through four residual blocks.

Table 7: Bottom Prior Conditioning Stack

| Parameter | Values |
|---|---|
| Input size | $32{\times}32{\times}4$, $64{\times}64{\times}4$ |
| Hidden units | 1024 |
| Residual Blocks | 20 |

Table 8: Let $n$ be the timestep to be modeled by the bottom prior. The $32{\times}32{\times}4$ top layer is upsampled to $64{\times}64{\times}8{\times}1024$ through a series of three convolutional layers with kernel sizes (4, 3, 3)$\rightarrow$(3,4,4)$\rightarrow$(3, 3, 3) and strides (2, 1, 1)$\rightarrow$(1, 2, 2) $\rightarrow$(1, 1, 1) respectively. From this output, the $n$th slice is chosen as input. For the bottom layer, an input the past $n-4, n-3, ...n-1$ timesteps are fed into a series of 4 convolutional layers at size (4, 3, 3) and stride (2, 1, 1) each. This downsamples to a layer of size $64{\times}64$. This is concatenated with the output from the top layer and fed into a conditioning stack identical to the one described in (Razavi et al., 2019)

