# OpenReview forum: "Predicting Video with VQVAE"
_ICLR.cc/2021/Conference — Reject_

### Official Review · AnonReviewer1 · 2020-10-27
**Not good enough**

**Rating:** 4
**Confidence:** 4

**Review:**

Summary:
This paper introduces a model combining VQ-VAE and autoregressive generative model for video prediction. The multi-level discrete latent variables help to predict higher resolution videos. The experiment on Kinetics-600 shows that the the model is able to predict 256x256 resolution videos.
-------------------------------------------------
Pros:
+ High resolution video prediction: I believe there is no existing video prediction model that predicts high resolution videos like 256x256.
+ The paper covers most of related references.
-------------------------------------------------
Cons:
1. *Lack of contribution.*
Hierarchical VAE is not new in the field [1,2]. The proposed approach combines two existing method (Pixel CNNs and VQ-VAE) for higher-resolution video prediction. Although, this paper claims that the model is able to predict higher-resolution video outputs, the quality of the prediction is not clearly evaluated in the paper. Regarding the quality evaluation, see Cons 3. for details.

2. *Missing reference* in terms of high-resolution video prediction: [3]

3. *Limited evaluation and comparisons.*
The introduced approach is evaluated on one dataset (Kinetics-600) using FVD and a human study. Other works, such as [Luc20], [Clark20], and [Weissenborn20], have shown prediction results on multiple datasets using multiple metrics.
For instance, UCF-101 is used by [Luc20] and [Clark20], and the BAIR robot pushing dataset is used by [Weissenborn20]. The KITTI driving dataset and Human 3.6M datasets are used by [1] for higher resolution video prediction.
I understand that Kinetics-600 contains many and diverse videos, and some of previously used datasets are in small resolution such as the BAIR robot pushing dataset. However,
    1. to understand the capability of the proposed approach, evaluating on more datasets is necessary such as the KITTI driving dataset or UCF-101.
    2. In addition, there is no best way so far to quantitatively evaluate the video quality with one measure. I suggest to provide evaluation with more metrics. SSIM and LPIPS are other common metrics together with FVD.
    3. It is not easy to judge the output quality by just looking at the prediction in Figure 4. Adding the ground truth next to the prediction output would be very helpful.
    4. Also, providing more output videos in appendix/supplementary  materials will be useful.
-------------------------------------------------
Minor comments:
- Table 2 caption needs a description of FVD*.
-------------------------------------------------
[1] Klushyn, et al., Learning Hierarchical Priors in VAEs, NeurIPS 2019.
[2] Zhao, et al., Learning Hierarchical Features from Generative Models, ICML 2017.
[3] Villegas, et al., High Fidelity Video Prediction with Large Stochastic Recurrent Neural Networks, NeurIPS 2019.

---

> ### Author Response · Authors · 2020-11-23
> **Response**
>
> We emphasize that VQVAE is distinct from traditional VAEs. VAEs often assume a prior with a continuous probability distribution, while VQ-VAE assumes a discrete one. A discrete latent representation of the input space makes the generation problem more tractable.
>
> We agree that [3] has tackled high-resolution video prediction and will include this citation; however they applied their model to highly constrained datasets such as Human3.6, KITTI, and Robot Pushing Datasets. High resolution models have never scaled to complex, unconstrained video datasets.
>
> Evaluation of video prediction and generative models broadly is still an open problem. We believe that the FVD has flaws;  nevertheless we report it given its established presence in prior work.  The FVD focuses on high frequency details that humans often do not notice. Prior work [4] demonstrates that classifier-based metrics such as FVD harm VQ-based models which discard high-frequency information via lossy compression. Another route could be log-likelihood. Given that one of our main goals is sample quality, log-likelihood metrics are not great for evaluating and comparing models [5]. This would be especially problematic given that many of our baselines are GANs. The last route could be PSNR or SSIM on the predicted frames, but these metrics are not suitable for multi-modal problems such as video prediction. Using PSNR or SSIM for model selection would lead to averaging over all possible outcomes, leading to blurry predictions. A perceptual metric such as LPIPS does not incorporate this stochasticity either. These issues motivate us to include a human evaluation in our submission given that human-perceived sample quality is our main goal. We can add more examples in the appendix. We can also add the ground-truth next to the prediction output, but we are not sure how informative this would be. We emphasize that the problem of video prediction is not deterministic; the goal is to learn an accurate distribution of multiple future outcomes in the video.
>
> [4]Razavi, A. van den Oord, A. and Vinyals, O. Generating Diverse High-Fidelity Images with VQ-VAE-2. NeurIPS 2019.
>
> [5]Theis, L. van den Oord, A., and Bethge, M. A. Note on the Evaluation of Generative Models. ICLR (2016).

---

### Official Review · AnonReviewer3 · 2020-10-27
**Incremental Novelty and Weak Evaluation**

**Rating:** 3
**Confidence:** 5

**Review:**

Summary:
This paper proposed a new approach for the video prediction task. The proposed model is built upon VQ-VAE, which has shown promising image generation qualities. The authors proposed to use VQ-VAE to perform compression (dimensionality reduction), which in return allows them to perform video prediction for high-resolution videos -- 256*256.

Pros:
1. The proposed framework is valid, and the equations in the paper describing the model are correct. There is no fundamental error.
2. The paper is well-written and easy to understand.
3. Qualitative results are promising.

Cons:
1. The biggest issue of this paper is the lack of novelty. In general, this paper is an incremental work based on VQ-VAE. It is a novel application, but the theoretical novelty is minimal.
2. The second major issue of this paper is that the quality of the evaluation is quite weak. Overall, there are two tables (table 1 and table 2) and one figure. Table 1 provides a human evaluation of the proposed Video VQ-VAE with baseline models. However, only one baseline model is compared.
3. The caption of table 5 and 7 are given as table 6 and table 8, which makes it very confusing to understand the content.
4. In section 5.2, the author mentioned that FVD results for full-resolution as a baseline are given in Table 7. I do not think this table is included in the paper.
5. Since the proposed model is built upon autoregressive models and VAE models, one important evaluation metric would be likelihood (or ELBO). There have been quite a few video prediction papers based on VAE models (such as "Probabilistic Video Generation Using Holistic Attribute Control" from ECCV2018), a thorough comparison with these model would provide more insight in the effectiveness of the proposed model.
6. Compressing data with latents is listed as an important novelty of the model. However, I am not convinced this is the case. Any embedding, whether it's a probabilistic encoder in VAE or deterministic encoder, will have the compressing effect. Actually, almost all of the previous time-series papers rely on the dimensionality reduction step.
7. In section 3.2, the authors claim that "the power-of-two design of our architecture leads us to condition on 4 and predict 12," while previous models condition on 5 and predict 11. This is listed as one of the benefits of the proposed model. However, most of the time-series models, no matter whether they are sequential latent variable models, or autoregressive models, or even simply an LSTM model, it is not a hard constrain on how many steps they can predict into the future. Theoretically, they can all predict infinite future steps. It is true that the quality of the prediction will decrease over time. But if the authors want to show that the Video VQ-VAE is superior from this perspective, experimental comparisons need to be provided to back up this statement.

---

> ### Author Response · Authors · 2020-11-23
> **Response**
>
> The full-resolution results refer to the  “Video VQ-VAE FVD* (256×256)” line on Table 2. We compare three baseline models with FVD via Table 2. Table 1 utilizes human evaluation vs the state of the art baseline given the flaws in FVD. As with our response to reviewer 1,  log-likelihood is another possibility. However, log-likelihood metrics are not great for evaluating models based on sample quality [5]. We note that our compression approach optimizes lossy compression rather than log-likelihood directly. Compression also discretizes the latent space making it more tractable for generative models. Our model is fully convolutional and can be theoretically extended to arbitrarily large timesteps; we clarify due to the architecture this is in multiples of 4 frames (i.e. 4, 8,12,16, 20, etc.)
>
> [5]Theis, L. van den Oord, A., and Bethge, M. A. Note on the Evaluation of Generative Models. ICLR (2016).

---

### Official Review · AnonReviewer2 · 2020-10-28
**Review for Predicting Video with VQVAE**

**Rating:** 4
**Confidence:** 4

**Review:**

### SUMMARY

The authors propose to use a VQVAE-2 setup for video prediction. In particular, they propose a hierarchical discrete latent variable model that compresses videos into a latent space. An autoregressive model is then used to model dynamics in this latent space, which has reduced dimensionality, and can be used together with the VQVAE decoder to predict video.  Empirical results show that this model is comparable to SOTA GAN models and a human evaluation suggests that humans have a preference for the
VQVAE generations.

### STRENGTHS AND WEAKNESSES

[+] Good empirical results
[-] Reduced novelty
[-] Weak empirical section - only one dataset, limited comparison to baselines, missing details
[-] Some claims are not properly justified


### DETAILED COMMENTS

The main positive aspect of the paper are its results. The authors show that, for the Kinetics-600 dataset, their approach performs similarly to SOTA GAN methods based on the FVD metric. Further, they conduct a human evaluation that indicates that their generations might be preferable to those from GANs.

However, this empirical evaluation is quite limited. The authors only show results for the Kinetics-600 dataset. It is true that this is one of the largest scale video datasets and that it is quite challenging. However, the authors could have shown results for other datasets (UCF101, Kitti/Cityscapes/BDD100K, etc.) that offer different trade-offs between complexity and amount of available data (note that for example BDD100K is in the same order of magnitude as Kinetics in terms of data available) and show the performance of their method on those datasets. Furthermore, that would allow a more direct comparison to some previous baselines including non-GAN approaches. The authors could also show different metrics beyond FVD)to assess the performance of their method - video prediction evaluation is an open research question and some metrics are known to have shortcomings, but given that there is no ideal metric then having multiple metrics could help have a better understanding on the model strengths and weaknesses.

Another issue is the missing details in terms of architectural choices, optimization hyperparameters, computational requirements, training time, etc. These are important to be able to replicate the results and to assess the potential impact of the method, as many video generation/prediction models have large computational requirements that have prevented a wider adoption of some of these methods.

One of the main shortcomings of the paper is its novelty. The paper is a straightforward application of the VQVAE2 model to 4D tensors (video) and for video prediction. The original VQVAE paper already shows some results on video, contrary to the claim that "this is the first application of VQVAE to video data". Further, using autoregressive models for video is a common approach (Video Pixel Networks, Weissenborn et al. 2020), and using an autoregressive model on the latent space of a discrete latent variable model is common in all VQVAE approaches. There is merit in showing that this setup works for video prediction, but nothing about the setup is novel.

Finally, there are some claims in the paper that are factually or arguably incorrect. These have not directly influenced my score, but I believe they should be addressed should the paper be accepted. First the authors claim "we know of no attempt to predict video at resolutions beyond 64x64, except for flow models". Clark et al., Castrejon et al. and even the codebase for Denton and Fergus, Lee et al. all have results/models for video prediction at 128x128.  Second, the authors claim that most video prediction methods have skewed towards using GANs. This is arguably not true, I believe most GAN models perform video *generation* (Vondrick, Tulyakov) and most current video *prediction* models are based on VAEs (Denton, Babaeizadeh, Lee, Castrejon, Villegas, etc.). This is due to the fact that usually it is hard to get GANs to cover all modes and usually they perform better for unconstrained generation rather that conditional generation/prediction. In any case, this is debatable and as such it should be clarified or toned down in the paper. Some of the other unjustified claims include "this model being the first application of VQVAE to video data" as discussed.


### SCORE

Overall I think this is a borderline paper. I think the empirical results are good and might be of interest to the community. However, the paper has important shortcomings - it has reduced novelty, there are important missing details, and the empirical evaluation is weak. Therefore in its current form I vote for rejection.

### POST-REBUTTAL UPDATE
The authors' response did not address my concerns. Given that most current evaluations metrics for video generation/prediction have some shortcomings (including human evaluations), it makes more sense to include a wide range of metrics that showcase the strengths and weaknesses of a method rather than to argue against their inclusion in the paper. Additionally, the authors failed to mention very relevant prior work (Latent Video Transformer).

Therefore I do not think this submission in its current form should be accepted and I have reduced my rating to a 4.

---

> ### Author Response · Authors · 2020-11-23
> **Response**
>
> We note that the video experiment in the original VQVAE paper was applied to sequences of static frames and did not explore the role of compression in the temporal (motion) dimension. We will clarify our claims in the paper, noting the importance to move video prediction models beyond constrained datasets such as BAIR, KITTI, and UCF101. We will also add more details in the appendix to facilitate reproducibility. We used two metrics in our evaluation - FVD and human evaluation - due to the fact that metrics such as PSNR/SSIM average over possible outcomes, and even log-likelihood [5] fall short of reflecting sample quality. We argue that given the flaws in current evaluation metrics, human evaluation is the best judge of sample quality.
>
> 5]Theis, L. van den Oord, A., and Bethge, M. A. Note on the Evaluation of Generative Models. ICLR (2016).

---

### Official Review · AnonReviewer4 · 2020-10-29
**Official Blind Review #4**

**Rating:** 4
**Confidence:** 3

**Review:**

Summary

The paper proposes a modification of hierarchical VQ-VAE for video prediction task. To model temporal dependency, the encoder and pixelCNN in original VQ-VAE are extended with 3D convolutions. The proposed method is evaluated on a large-scale video dataset, Kinetics-600 dataset.

Pros
- The paper is generally well-written and easy to follow.
- The idea of employing hierarchical and quantized representation to model complex variations in video sounds reasonable. The overall algorithm seems to be a reasonable extension of VQ-VAE to videos.

Concerns & Suggestions
- Although the idea is reasonable, the paper simply extends the hierarchical VQ-VAE for images to videos with modification to model temporal dependency using 3D convolutions. Thus, the technical contribution of the paper is limited.
- The paper inherits the pixel-wise autoregressive model for modeling prior distribution. However, the pixelCNN is based on highly serialized computations, and generally suffers from a very long inference time. It would be interesting to see the analysis of inference time and some justifications that pixelCNN fits the video prediction at a large-scale.
- The arguments on worse FVD performance require further justifications; it is unclear why the likelihood-based method has a disadvantage over the adversarial methods in terms of FVD, as the FVD is a model-neutral metric.
- The paper presents a human study for qualitative comparisons. However, the number of both participants and presented videos are too small to draw meaningful conclusions.

--- post rebuttal update ----

I appreciate the authors for their response. However, the arguments on FVD are still not quite convincing (i.e., FVD still has reasonable correlation with actual generation quality; if the perceptual metric is inappropriate, then the authors should have tried other metrics. Also, the authors did not address other concerns such as concerns on computation time, scalability, small-scale human evaluation, etc. I maintain my score to rejection of this paper.

---

> ### Author Response · Authors · 2020-11-23
> **Response**
>
> We note that the compressed latent space from VQ-VAE allows us to scale autoregressive models to much higher resolutions than before (i.e. Video Transformer only scales to 64x64). As with our response to Reviewer 1 and 3, We note that FVD focuses on high frequency details that humans often do not notice. Prior work [4] demonstrates that classifier-based metrics such as FVD harm VQ-based models which discard high-frequency information via lossy compression. Log-likelihood metrics are also problematic as they do not reflect sample quality [5]. This would be especially problematic given that many of our baselines are GANs.
>
> [4]Razavi, A. van den Oord, A. and Vinyals, O. Generating Diverse High-Fidelity Images with VQ-VAE-2. NeurIPS 2019.
>
> [5]Theis, L. van den Oord, A., and Bethge, M. A. Note on the Evaluation of Generative Models. ICLR (2016).

---

### Public Comment · ~Denis_Volkhonskiy1 · 2020-11-12
**Previous work with VQVAE for videos**

The idea of using VQVAE for videos is not novel. Previously proposed Latent Video Transformer (https://arxiv.org/pdf/2006.10704.pdf) also consider using VQVAE and recurrent model for predictions in the latent space. It would be nice to compare your work with it in terms of computational requirements. How many GPUs did you use for training comparing to 8 GPUs in Latent Video Transformer?

---

> ### Author Response · Authors · 2020-11-23
> **Response**
>
> We recognize Latent Video Transformer as related, concurrent work in this space and will include a citation and clarify the differences of this model in the paper. However, we also emphasize that our approach is the first VQVAE based approach to include hierarchy to scale to high resolutions.

---

### Decision · Program_Chairs · 2021-01-07
**Final Decision**

**Decision:**

Reject

**Comment:**

While this paper was perceived as being fairly well written, the level of novelty and the evaluation were seen as weak by many reviewers. The aggregate opinions across reviewers is just too low to warrant an acceptance rating by the AC. The AC recommends rejection.